# Wheat, Barley, and Oat Breeding for Health Benefit Components in Grain

**DOI:** 10.3390/plants10010086

**Published:** 2021-01-03

**Authors:** Igor G. Loskutov, Elena K. Khlestkina

**Affiliations:** Federal Research Center the N.I. Vavilov All-Russian Institute of Plant Genetic Resources (VIR), St. Petersburg 190000, Russia; e.khlestkina@vir.nw.ru

**Keywords:** barley, breeding, marker-assisted selection, genes, genetic resources, genome editing, health benefits, metabolomics, oat, QTL, wheat

## Abstract

Cereal grains provide half of the calories consumed by humans. In addition, they contain important compounds beneficial for health. During the last years, a broad spectrum of new cereal grain-derived products for dietary purposes emerged on the global food market. Special breeding programs aimed at cultivars utilizable for these new products have been launched for both the main sources of staple foods (such as rice, wheat, and maize) and other cereal crops (oat, barley, sorghum, millet, etc.). The breeding paradigm has been switched from traditional grain quality indicators (for example, high breadmaking quality and protein content for common wheat or content of protein, lysine, and starch for barley and oat) to more specialized ones (high content of bioactive compounds, vitamins, dietary fibers, and oils, etc.). To enrich cereal grain with functional components while growing plants in contrast to the post-harvesting improvement of staple foods with natural and synthetic additives, the new breeding programs need a source of genes for the improvement of the content of health benefit components in grain. The current review aims to consider current trends and achievements in wheat, barley, and oat breeding for health-benefiting components. The sources of these valuable genes are plant genetic resources deposited in genebanks: landraces, rare crop species, or even wild relatives of cultivated plants. Traditional plant breeding approaches supplemented with marker-assisted selection and genetic editing, as well as high-throughput chemotyping techniques, are exploited to speed up the breeding for the desired genotуpes. Biochemical and genetic bases for the enrichment of the grain of modern cereal crop cultivars with micronutrients, oils, phenolics, and other compounds are discussed, and certain cases of contributions to special health-improving diets are summarized. Correlations between the content of certain bioactive compounds and the resistance to diseases or tolerance to certain abiotic stressors suggest that breeding programs aimed at raising the levels of health-benefiting components in cereal grain might at the same time match the task of developing cultivars adapted to unfavorable environmental conditions.

## 1. Introduction

Cereal crops are the main food and feed sources worldwide, supplying more than half of the calories consumed by humans [1]. An overwhelming majority of plant breeders and geneticists work on no other crops but cereals. Breeding methods depend on the biological features of a crop and on the genetic research standards, traditions, economic objectives, and levels of agricultural technologies in the country where plant breeding is underway. The general breeding trend of the past decades, however, was finding solutions to the problem of higher yields in cereal crops; furthermore, special attention was paid in many countries to increasing plant resistance against diseases and various abiotic stressors. The concentration of all efforts on these two targets and none other resulted in a certain decline in the genetic diversity in those plant characters that are associated with the biochemical composition of cereal grain [2]. In the last few years, cereal crop breeding generated a trend aimed at combining high biochemical and agronomic parameters in one cultivar [3,4,5]. In addition to protein, cereal grains are rich in other chemical compounds, such as fats with their good assimilability by the organism and a well-balanced composition of chemical constituents, including fatty acids [6,7,8,9,10], vitamins of the В, А, Е, and F groups, organic compounds of iron, calcium, phosphorus, manganese, copper, molybdenum, and other trace elements [3], and diverse biologically active compounds–polysaccharides, phenolic compounds, carotenoids, tocopherols, avenanthramides, etc.

In recent years, the world food market has seen the emergence of a wide range of new cereal crop products designed for dietetic purposes. Currently, available data confirm the importance of biochemical composition in cereal crop grains since it underpins their dietetic, prophylactic, and curative effect on the human organism [11]. Cereals are rich in protein, starch, oils, vitamins, micronutrients, and various antioxidants. The research that examines the potential of a number of cereal crops for prophylactic or medicinal uses has been expanding from year to year [12,13,14,15,16]. In addition to determining types of bioactivity for different grain components, an important challenge is to concentrate further efforts of researchers on disclosing the mechanisms of their effect [17].

It is admitted that breeding techniques can help to increase the percentage of individual constituents in the grain to a very high level. An important role in promoting this breeding trend is played by the achievements in modern genetics of cereal crops and traits associated with the quality and dietary value of their products. New breeding programs imply that the developed high-yielding cultivars will combine maximum contents of the abovementioned components and optimal correlations among them with other grain quality indicators and resistance to biotic stressors. Marker-assisted selection techniques are used more and more often to accelerate the development of cultivars enriched in useful grain components [4,18]. There are examples of the works employing genetic editing technologies for these purposes [19,20,21]. The current review aims to consider current trends and achievements in wheat, barley, and oat breeding for health-benefiting components.

## 2. Major Dietary Components in Grain and Breeding Programs for Health Benefit

### 2.1. Micronutrients

The long-standing problem of micronutrient deficiencies in human diets is the most significant for public healthcare worldwide. It is especially true for cereal-based diets: They are poor in both the number of micronutrients and their bioavailability for the organism since breeding of these major food and feed crops primarily aims at developing higher-yielding varieties to meet global demand. Due to dilution effects, an increase in grain mass sometimes causes a reduction in micronutrient contents. In most countries, people eat meals produced from cereal crops with low micronutrient content; it is a serious global problem invoked by the uniformity of different diets and may lead to significant health deteriorations [22,23]. Iron-deficiency anemia is one of the most widespread health disorders provoked by the worldwide deficit in micronutrients [24], while zinc deficiency in food is faced on average by one-third of the world’s population [25]. Increasing the content of these trace elements in wheat by breeding techniques is considered one of the ways to enhance the consumption of micronutrients with food [26].

It has been noticed that cereal crop cultivars can be enriched in the desired micronutrients through the application of agricultural practices or by plant breeding [22,27,28,29,30]. Such procedures, however, might lead to an increase in micronutrient content in leaves but not in grain [31]. Methods combining breeding and agrochemical approaches were proposed to solve this problem: They helped accumulate micronutrients in the edible parts of plants [27,28,29,32]. There are considerable variations in the concentration of micronutrients in seeds or kernels of most crops [3,32]. Genetic variability in the micronutrient content is often observed to be less expressed in fruit and more in leaves. Nevertheless, screening large collections of staple cereal crops reveals extensive diversity of micronutrient concentrations in their grains [26,32,33]. Increased content of most micronutrients was observed in local varieties and landraces of wheat and other cereals, compared with improved commercial cultivars [34].

The content of micronutrients in grain was analyzed in 65 commercial Russian cultivars of four major cereal crops: wheat, barley, rye, and oat. Statistically significant variations were found in the content of all studied trace elements (Fe, Zn, and Mn). The highest levels were registered for barley and oat cultivars. Among barley genotypes, the content of Fe, Zn, and Mn varied with a 3-to 5.5-fold difference between the extremes (Table 1). Oat cultivars manifested a 7-fold difference between the extremes in the Zn content and nearly 3-fold in Mn [3].

A detailed study of a set of commercial oat cultivars of different geographical origin in the context of their micronutrient content and biochemical parameters showed that genotypic differences in the Fe and Zn levels in grain were small (1.9–2.7 times), but in Mn, they were relatively high (10.5 times). A 1.8-fold difference was observed between the lowest (10.9%) and the highest (19.3%) protein content levels in oat grain [3]. A wide range of variation in oil content (2.7–8.1%) was found in all studied oat accessions. The amounts of protein, oil, oleic acid, and Zn in grain demonstrated statistically significant positive correlations among themselves [3]. The identified oat cultivars with high nutritive value will be included in breeding programs and used directly in high-quality food production.

Molecular-genetic research on 335 spring barley accessions was conducted for more effective utilization of the micronutrient diversity in cereal crop breeding. A genome-wide association study (GWAS) was employed for mapping quantitative trait loci (QTL) linked to the content of macro- and micronutrients in grain (Fe, Zn, Ba, Ca, Cu, K, Mg, Mn, Na, P, S, Si, and Sr). The analyses of the tested populations helped to identify specific QTL for each of the studied indicators and map them on chromosomes. The QTL identified are valuable for the future development of barley cultivars with increased content of nutrients, especially Zn and Fe [35].

### 2.2. β-glucans

A physiologically important dietary component in the grain is (1,3;1,4)-β-D-glucan, or the non-starchy water-soluble polysaccharide β-glucan. This component is reported to be typical of some species of the Poaceae family: its content varies within 3–11% in barley, 1–2% in rye, and <1% in wheat, while in other cereals, it is present only in trace amounts [36]. At the same time, the content and composition of dietary fibers in various cereal crop species are genetically determined. It means, as opined by many scientists, that it is possible to produce new lines of such crops with different correlations between the levels of β-glucan polysaccharides and arabinoxylans that would be optimal for various uses [37,38,39]. Studying of the β-glucan content in oat and barley cultivars is associated with their uses for dietetic and medical purposes [37,38].

The β-glucans are not evenly distributed within a grain: its larger amount is found in the endosperm cell walls, aleurone, and subaleurone layers, and its content varies from 1.8 to 7% [40,41]. The concentration of β-glucans in oat grain and their degree of polymerization depend not only on the cultivar but also on the conditions of cultivation, grain processing, and post-harvest storage [42].

The presence in the grain of a higher amount of β-glucans, which are dietary soluble fiber (or soluble non-starch polysaccharide), determines the viscosity of oat and barley broths, which have a beneficial effect on important functions of the human gastrointestinal tract, so they are widely used in the food industry for dietetic and curative purposes [36,43]. Among numerous products of barley and oat biosynthesis, probably the most valuable for the human organism is soluble cellulose fibers and β-glucans first of all (also arabinoxylan, xyloglucan, and some other secondary cellulose components), as they can reduce the level of cholesterol in the blood and noticeably mitigate the risk of cardiovascular diseases [38,44,45]. Multiple evidence of the beneficial role played by β-glucans impelled the U.S. Food and Drug Administration (FDA) to make an official statement that soluble dietary fibers extracted from whole oat grain to produce flakes, bran, or flour helped to reduce the risks of cardiovascular diseases [46]. Insoluble fractions of dietary fiber are partly cellulose, xylose, and arabinose [39]. Insoluble dietary fiber has general gastrointestinal effects and, in most cases, has an impact on weight loss. There is convincing evidence that β-glucans contained in oat grain are partially responsible for decreasing the levels of glucose in the human blood and of cholesterol in serum [12]; it is associated with its physicochemical and rheological characteristics, such as molecular weight, solubility in water, and a viscosity [42,47].

Genetic diversity of barley and oats in the content of β-glucans in their grain was evaluated in the framework of two European Union (EU) programs. The HEALTHGRAIN Diversity Screen project resulted in finding significant differences in the content of β-glucans and antioxidants in the grain of five tested oat cultivars [48]. The AVEQ project (*Avena* genetic resources for quality in human consumption) analyzed 658 oat cultivars and confirmed the contribution of both genetic and environmental aspects to the formation of the tested character [49]. It is worth mentioning that, compared with cultivated and other wild di- and tetraploid oat species, higher contents of β-glucans and other antioxidants were found in the hexaploid (wild) *A. fatua, A. occidentalis,* and (cultivated) *A. byzantina*, and diploid (wild|) *A. atlantica* [38,39,49,50,51].

Measuring the content of β-glucans in oat grain in large and diverse sets of cultivars and species showed that its values were significantly dispersed [37,38,49]. Naked oat forms demonstrated a higher total content of the analyzed polysaccharide than hulled ones, but the latter contained more insoluble β-glucans in their grain [52,53,54]. Computer modeling helped to provide a ranking of the factors affecting the β-glucan content in hulled and naked oat cultivars during their cultivation. The analysis showed that the selection of the cultivar is the most important parameter of the model for determining the final β-glucan accumulation in grain, among the other factors [55]. There are contradictory data concerning the results of comparative studies on naked and hulled barley as well. Some authors failed to disclose significant differences between these two forms of the crop [56,57], while others found that naked barleys contained more β-glucans than hulled ones [43,58]. Meanwhile, the group of Tibetan naked barleys was reported to have the highest content of β-glucans in their grain [59].

In the meantime, the amount of β-glucans in oat grain is associated with protein and fat accumulation, grain volume weight, and grain productivity [60,61]. The content of these polysaccharides depended on meteorological conditions and agricultural practices used in oat cultivation [61]. The content of β-glucans in barley grain is determined by both the genotype and the growing conditions [43,59,62]; some authors insist that it is the genotype that plays a decisive role [63,64], while others give preference to the environmental conditions [65,66]. When 33 barley cultivars and lines were tested in two arid areas in the United States, it was shown that the variability in the content of β-glucans in grain was determined by the genotype for 51% [64] to 66% [67]. At the same time, the protein content in grain depended on environmental conditions for 69%, whereas yield size and the grain volume weight for 83 and 70%, respectively [64]. The study of 9 barley cultivars and 10 oat ones showed that cultivar-specific differences in the β-glucan content persisted across the years [63]. The content of β-glucans in the grain is also influenced by plant development phases. It was reported that the content of β-glucans gradually increased in the process of grain formation, and in the maturation phase, it either reached the plateau or decreased [57]. At present, there are contradictory data concerning the linkage of β-glucan accumulation in barley grain with 1000 grain weight, protein content, or starch content [56,62]. Some authors did not find any interplay between these characters, while others reported a positive correlation. When the content of β-glucans was measured in the grain of six-row and two-row barley cultivars, no differences between these two cultivar groups were reported [43].

The 1700 oat lines with mutations induced by TILLING of high-frequency mutagenesis have been produced for breeding purposes with molecular-based, high precision selection methods from cv. ‘Belinda’ (Sweden) to evaluate the variability of β-glucans content in this crop [68]. Their assessment resulted in identifying 10 lines with β-glucan concentrations in their grain higher than 6.7% and 10 lines with the content of β-glucans less than 3.6% (β-glucan concentration in cv. ‘Belinda’ was 4.9%). The maximum range of variation in the content of these polysaccharides was from 1.8 to 7.5% [69]. The comparatively recent identification of genes participating in the biosynthesis of β-glucans in cereals [70] and their first genetic map open new opportunities for genetic improvement of grain quality indicators and resulting food products, which is very important for human health [71].

Three markers (Adh8, ABG019, and Bmy2) significantly linked to β-glucan content regulation were identified in barley grain, and a group of *HvCslF* genes was mapped: At least two of them were in the region of barley chromosome 2H explained by the QTL for (1,3;1,4)-β-glucan near the Bmy2 marker [72]. A genome-wide association study (GWAS) employing oat germplasm of worldwide origin from the American Gene Bank was aimed at the identification of QTL linked to β-glucan content in grain and resulted in finding three independent markers closely associated with the target character. A comparison of these results with the data obtained for rice showed that one of the described markers, localized on rice chromosome 7, was adjacent to the *CslF* gene family responsible for β-glucan synthesis in grain. Thus, GWAS in oat can be a successful QTL detection technique with the future development of higher-density markers [73].

By now, the GWAS approach has already started to be used to analyze the association between the genotype and the content of β-glucans and fatty acids in oats. Researchers have identified four loci contributing to changes in the fatty acid composition and content in oat grain. However, genome regions conducive to changing the content of proteins, oils, saccharic and uronic acids, which, in their turn, produce a direct effect on grain quality, remain unexplored [74]. Furthermore, positive correlations were demonstrated in barley between 1000 grain weight and tocol concentration, between dietary fiber content and phenolic compounds, and between husk weight and total antioxidants in hulled barley [38,50].

### 2.3. Antioxidants

Cereal crop grains are known to have high nutritive value and contain diverse chemical compounds with antioxidant properties. Research efforts have been undertaken in recent years to study the content of antioxidants in the grain of various cultivated cereals [50,75,76,77,78,79].

Starting in the mid-1930s, oat flour has been used as a natural antioxidant. Later, more in-depth research was done to assess the antioxidative properties of oat flour versus those of chemical antioxidants. It was ascertained that adding sterols extracted from oat to heated soybean oil significantly decelerated its oxidation compared with the reference. At present, along with the extensive utilization of synthetic antioxidants, oat flour has found its stable niche as a natural ingredient in eco-friendly food products [7].

A comparison of bakery products made from wheat that synthesized such antioxidant compounds as anthocyanins with those from an anthocyanin-free wheat line demonstrated that the presence of anthocyanins increased the shelf life of bakery products and their resistance to molding under provocative conditions [80]. Cereal crops contain secondary metabolites with antioxidant activity belonging to three groups: phenolic compounds, carotenoids, and tocopherols [81].

### 2.4. Phenolic Compounds and Avenanthramides

Oat and barley grains contain a considerable amount of various phenolic compounds exhibiting biological activity, including antioxidative, anti-inflammatory, and antiproliferative (preventive activity against cancerous and cardiac diseases) effects [50]. One of the most abundant and powerful antioxidants found in nature, the flavonoid quercetin, has been found in wheat. It is characterized by numerous biological effects, including antithrombotic activity [82].

Many published studies testify that a major part of phenolic compounds in grain occurs in a bound form: Their content in oat and wheat grains reaches 75% [83,84]. Phenolic acids, like most flavonoids in cereal crops, are concentrated in structures bound to the cell wall: 93% of the total flavonoid content in wheat and 61% in oats [83]. The highest level of total flavonoids is characteristic of maize grain, followed by wheat, oats, and rice [83]. Phenolic acids are the most widespread phenolic compounds in oats, especially ferulic acid (250 mg/kg), which is present mainly inbound forms linked through ester or ether bonds to cell wall components but also exists in the free form [85].

Bioactive chemical compounds are unevenly distributed within the grain. Grains of four naked barley cultivars were divided into five layers to measure the total phenolic content and total antioxidant activity. The total content of soluble phenolic compounds was observed to decrease from the outer layer (2.8–7.7 μg/g) towards the inner endosperm structures (0.87–1.35 μg/g) [78,86]. It has been proven that most antioxidants contained in whole grain are located in the bran and germ fractions of the grain. For example, whole-grain wheat flour was found to contain in its bran/germ fraction 83% of the total phenolic content in grain and 79% of total flavonoids [87].

In the study of molecular mechanisms of ‘melanin-like’ black seed pigments known to be strong antioxidants, comparative transcriptome analysis of two near-isogenic lines differing by the allelic state of the *Blp* (black lemma and pericarp) locus revealed that black seed color is related to the increased level of ferulic acid and other phenolic compounds [88]. The melanic nature of the purified black pigments was confirmed by a series of solubility tests and Fourier transform infrared spectroscopy, while intracellular pigmented structures were described to appear in chloroplast-derived plastids designated “melanoplasts” [89]. The most frequently mentioned flavonoids of cereal crops are the flavonols kaempferol and quercetin, the flavanone naringenin and its glycosylated forms, catechin, and epicatechin in barley [90,91,92,93].

Pigmentation of the grain’s outer coating can be analyzed as an important indicator of antioxidant activity. A barley cultivar with purple grain contained 11 anthocyanins, while only one anthocyanin was observed in black and yellow barley grains. The purple barley bran extract had the highest total antioxidant activity [94]. Another study of naked barley demonstrated the presence of higher antioxidant activity in pigmented grains compared with non-pigmented ones [78]. A study of naked and hulled oats showed that naked oat cultivars had significantly higher values of total antioxidant activity. Among hulled oat cultivars, these values were higher in dark-hulled forms compared with white-hulled oats [50].

Differences between naked and hulled oats and barleys, generated a perfect model interesting for comparative analyses: the mutant barley line for the *Nud* gene (nakedness), derived by gene editing from cv. ‘Golden Promise’ [21]. Using this model will help to distinguish the pleiotropic effects of the *Nud* gene on the grain’s biochemical composition from the influence of closely linked genes.

Analyzing grain extracts of wheat lines with different combinations of the *Ba* (*Blue aleurone*) and *Pp* (*Purple pericarp*) genes on the genetic background of elite cultivars demonstrated a higher diversity of flavonoid compounds in the carriers of dominant alleles of *Ba* and *Pp* genes. Comparing the products made from the grain of a purple-grained line with those from an anthocyanin-free isogenic line revealed significant differences, which was also true for the samples that had passed a full processing cycle, including baking at elevated temperatures [80,95]. The analysis of anthocyanin extracts obtained under conditions simulating those of food digestion by a human organism showed that ingesting 100 g of bread crisps or biscuits made from flour with added purple wheat grain bran raised the assimilation of anthocyanins to 1.03 and 0.83 mg, respectively, i.e., 100 g of bran would supply the organism with up to 3.32 g of anthocyanins. Besides, purple-grained wheat matched or even exceeded the reference line in the quality and taste of its products [95].

Recently, new high-yielding wheat cultivars, resistant to fungal diseases and having high anthocyanin content in grain have been developed [4]. The efficiency of the breeding strategy lasting only three years from the first cross until the state cultivar competitive testing has been demonstrated. The strategy is based on marker-assisted selection (MAS) [4]. MAS also demonstrated its efficacy in creating barley with certain alleles of anthocyanin regulatory genes [18]. For breeding blue-grained wheat, besides molecular markers, FISH or C-banding are needed since the *Ba* gene is alien for wheat and can be inherited from wheat lines with either 4B or 4D chromosome substituted by the *Thinopyrum ponticum* chromosome 4 [96,97]. Unlike bread wheat, barley has its own *Ba* gene. Recent findings of regulatory features of anthocyanin biosynthesis in barley [98] are useful for both MAS-based and genetic editing-based breeding strategies.

Interestingly, 30 years ago, the purple- and blue-grain characters were regarded as having “a limited practical use from a scientific point” [99]. Since that time, some studies demonstrating the health benefit of plant anthocyanins, including those from wheat grain [16], have been carried out, denying the old point of view and proving these traits to be economically important. Commercial cultivars of wheat with increased anthocyanin content have been released in Canada, China, Japan, and several European countries [100,101].

The class of phenolics with antioxidative effect and bioactivity includes avenanthramides (AVA), a class of hydroxycinnamoyl anthranilate alkaloids contained only in oats. Twenty-five components of these compounds were detected in kernels, and twenty in hulls [102]. The most widespread in oats are AVA-A (2p), AVA-B (2f), and AVA-C (2c) [9,103,104]. There is documented evidence that avenanthramides demonstrate antioxidant, anti-inflammatory, antiatherogenic, and antiproliferative activity [105,106,107].

It has been shown that oat cultivars differed in the AVA content in grain. The cultivated diploid species *A. strigosa* had a very high AVA content reaching 4.1 g/kg, and the hexaploid *A. byzantina* contained 3.0 g/kg. Contrariwise, wild oat species with different ploidy levels were characterized by relatively low AVA content values (240–1585 mg/kg) [108]. Analyzing a representative set of cultivated and wild oat species revealed an even wider diversity of the AVA content in grain [109]. A conclusion has been made that wild oat species are an important source of diversity for breeding programs, which dictates the necessity of further studies into the pattern of AVA content and composition variability across the genus *Avena* L. Wild oat species might incorporate a unique AVA composition, promising for crosses with cultivated oats.

### 2.5. Tocols

The health benefits of oats are also associated with the presence of several antioxidant compounds known as tocols, specifically tocopherols and tocotrienols. The fat-soluble vitamin E contains tocopherols and tocotrienols [110], which make the oil more resistant to oxidation. Both tocopherols and tocotrienols have several isomeric forms designated as α, β, γ, and δ [111]. All in all, vitamin E can comprise eight isomers, with prevailing α-isomers (70–85%) and δ-isomers not exceeding 1%. The total tocopherol content in oat cultivars can reach 2.6–3.2 mg/100 g, which is many times lower than in barley [101]. Tocopherols are mainly present in the germ fraction of grain, while tocotrienols are found in the pericarp and endosperm. Tocotrienols prevail in oats, barley, and wheat; their concentrations vary from 40 to 60 μg/g depending on the crop [112].

Eight isomers of tocols have been found in barley grain oil (four tocopherols and four tocotrienols). They play an exceptionally important role, regulating cholesterol in human blood. Tocols also demonstrate very high activity as antioxidants, blocking harmful peroxidation of lipids in cell membranes [101]. Tocols (16–94 mg/kg) consist of a polar chromanol ring linked to an isoprenoid-derived hydrocarbon chain. They are powerful scavengers of free radicals, also demonstrating an ability to inhibit the proliferation of some cancer cells [108].

Furthermore, positive correlations were demonstrated in barley between 1000 grain weight and tocol concentration, between dietary fiber content and phenolic compounds, and between husk weight and total antioxidants in hulled barley [38,50]. Presently, molecular-genetic studies of this type of antioxidant are based on simple-sequence repeats (SSR) markers. It is worth mentioning that the naked barley with the *Waxy* gene and zero amylase content in starch has higher contents of both β-glucans and tocols [113].

### 2.6. Sterols

Sterols are important components of vegetable oils. Their content in oat grain varies, according to different sources, from 0.1% to 9.3% of the total fatty acid content. This indicator often depends not only on the oat genotype but also on the extraction technique. Сultivars of rye, wheat, barley, and oats grown in the same year and same location were compared, the highest plant sterol content was observed in rye (mean content 95.5 mg/100 g, wb), whereas the total sterol contents (mg/100 g, wb) of wheat, barley, and oats were 69.0, 76.1, and 44.7, respectively [114]. Among the six components of sterol content, the main one is sitosterol, whose content reaches 70% of the total sterol content; additionally, about 20% are allocated to campesterol and stigmasterol [7,101]. The content of sterols in oats can reach 447 mg/kg and include, in addition to the abovementioned, D-5 and D-7 avenasterols [114] and phytic acid (5.6–8.7 mg/g); the latter manifests antioxidant activity due to its ability to chelate metal ions, thus making them catalytically inactive and inhibiting the metal-mediated formation of free radicals. However, this chelating activity reduced the bioavailability of major minerals [110].

### 2.7. Carotenoids

Carotenoids (yellow, orange, and red pigments) relating to isoprenoids are among the most widespread plant antioxidants. Carotenoid content in oat grain can reach 1.8 μg/g [86]; besides, lutein is considered the main xanthophyll in wheat, barley, and oat grains, and zeaxanthin is the secondary one [115].

Comparative investigation of four groups of wheat genotypes (spelt wheat, landraces, old cultivars, and primitive wheat) for carotenoid content and composition in grain revealed a high level of variation among the genotypes and the groups in the content of carotenoids. Lutein contributed 70–90% of the carotenoids in the grain [116]. In durum wheat, which is used for the production of pasta, carotenoid content is also an important technological and market indicator. In semolina and pasta, a yellow color is desirable, and it depends on the carotenoid accumulation in kernels. Genetic dissection of the carotenoid content character showed quantitative trait loci (QTL) on all wheat chromosomes [117]. The major QTL, responsible for 60% of heritability, is located on the long arms of chromosomes 7A and 7B. Variability in these QTL is explained by allelic variations of the phytoene synthase (PSY) genes. Molecular markers for MAS-based breeding programs aimed at the enrichment of durum wheat grain with carotenoid content are available [117].

### 2.8. Other Antioxidant Compounds

Oat is the only cereal grass that contains saponins, steroidal glycosides known as avenocosides A and B (65.5 and 377.5 mg/kg, respectively), which exhibit anticancer activity at the expense of diverse, complex mechanisms, including inhibition of neoplasm cell growth through cell cycle arrest and, inter alia, stimulation of cancer cell apoptosis [13]. Oat also accommodates two classes of saponins: avenocosides (steroid-linked saccharides) and avenacides (triterpenoid-linked saccharides), which were shown to drop the cholesterol level, stimulate the immune system, and demonstrate anticancer properties [14]. Targeted breeding for increased content of these compounds in oat lines has not yet been attempted, but interline and interspecies differences in this indicator have already been identified [118]. Grains of five Finnish barley cultivars grown in 2006–2008 were analyzed for their total content of folic acid. It was noted that the external and germ-containing grain layers had the highest levels of this compound (up to 1710 ng/g) [77,79].

## 3. Assessment of Cereal Crop Genetic Resources According to the Diversity and Concentration of Health-Friendly Dietary Grain Components

Secondary metabolites associated with quality traits in the released and processed products are presently identified using metabolomic profiling or chemotyping. Such an approach enables researchers to evaluate plant genetic resources according to these traits, including varieties of cultivated species and populations of wild ones. Chemotyping the grain of cultivated and wild *Avena* L. spp. showed that the range of variability in the metabolomic profile of improved cultivars was significantly narrower than that of wild species. Metabolites, the content of which may have been reduced in the process of domestication and breeding in comparison to wild oats, are identified [2]. Presumably, it might be connected with the selection during oat domestication and a decline of metabolome diversity while “domestication syndrome” traits were shaped [119]. The diversity of metabolomic profiles may be lost in the process of selection when highly specialized single-line intensive-type cultivars are developed because this process is always accompanied by a decrease in genetic polymorphism in a breeding object compared with the metagenome of numerous ecotypes, local varieties, and natural races of dozens of wild species [2,119]. A study of naked and hulled oat forms disclosed differences in their metabolites, which serves as an additional justification of the differentiation between these subspecies of common oat [2]. Landraces, which are plant varieties selected and grown regionally but not officially tested and released as registered varieties, are a source of special genetic characteristics derived by (many years of) adaptation to the respective territory. Such local varieties are often more resistant to biotic and abiotic stresses typical for their environment. In addition, such varieties may be a source for special phytochemicals (also known as bioactives) considered as health-beneficial, while the content of these compounds may be lower in commercial cultivar [2,120].

The bands of secondary metabolites in oat accessions exposed to *Fusarium* infection were analyzed, and correlations between metabolites and resistance were disclosed. High-protein oat forms with increased content of certain secondary metabolites demonstrated less damage from *Fusarium*, accumulated fewer toxins, and were more adaptable to the biotic stress [121].

Matthews et al. [122] used metabolite profiling to compare 45 lines of tetraploid and hexaploid wheat. The extracts were analyzed by the ultraperformance liquid chromatography coupled with time-of-flight mass spectrometry (UPLC-TOF-MS). Two different species of bread and durum wheat formed two distinct groups differing in sterols, fatty acids, and phospholipids, while *T. aestivum* L. split into two groups (corresponding to hard and soft bread wheat) according to differences in heterocyclic amines and polyketides. This and similar studies underpin the use of chemotyping in breeding both for desired agronomic traits and for higher contents of health-benefiting compounds in cereal grain. 

Information obtained with the molecular metabolomic approach on mQTL (metabolite quantitative trait loci) and mGWAS (metabolome-based GWAS) ensures a new level for qualitative and quantitative characterization of secondary metabolites interesting for breeding. Such analyses can provide knowledge about the interactions among metabolites themselves and between them and important breeding indicators. It may lead to the development of more rational models linking a certain metabolite with such characters as plant productivity or end-product quality. Even more promising is the possibility to examine the interplay between quantitative variation in metabolites and changes in the plant phenotype [123].

Due to the genetic potential of grain crops through the directed formation of the properties and structure of the kernel in the process of ontogenesis, when developing new cultivars, it is possible to attend to the target component composition of the final product. Wider application of chemotyping, chemical research methods, metabolomic analysis of grain quality, and searching for high content of rare beneficial (dietary or curative) components will result in the release of new crop cultivars, thus promoting next-generation breeding trends and technologies [50].

## 4. The Effect of Dietary Components in Grain on Life Functions of Plants Themselves

Content of all biochemical components in the grain of cereal crops there are variations in the composition of it. These variations arise from differences between environments, variation in the genotype of the crop, and interactions between biotic and abiotic factors and genotype. Biotic and abiotic factors change depending on climate change, soil, and various stressors affecting plants. The genotypic variation includes the differences between individual genotypes.

### 4.1. Biotic Stress Resistance

Generally, an explanation why grain in the soil is not affected by microorganisms despite the environmental conditions favorable for infection was given by the presence of antimicrobial flavonoid compounds in extracts from barley and wheat grains soaked in water [124]. Higher disease resistance of plants with enhanced flavonoid biosynthesis has been described in rye, barley, and wheat [125]. In vitro infection of developing barley caryopses of wild type and proanthocyanidin-free mutants with fungal pathogens *Fusarium poae*, *F. culmorum,* and *F. graminearum* revealed mutants to be more sensitive to *Fusarium* attack than wild-type plants [126].

Considering the available data on interactions between compounds with antioxidant properties in cereal crop kernels and *Fusarium* spp., it seems appropriate to suppose that some of the former could significantly contribute to the grain’s protection mechanism against toxicogenic fungi and mycotoxin accumulation. It has been proven that the crucial role in Fusarium Head Blight (FHB) resistance is played by five main classes of antioxidant metabolites: phenolic acids, flavonoids, carotenoids, tocopherols, and benzoxazinoids [127].

Cereal crop diseases caused by pathogenic and toxicogenic species of the *Fusarium* genus (FHB) inflict serious economic losses worldwide. Therefore, the development of sustainable strategies to prevent FHB contamination and mycotoxin accumulation has become a target of intensive research in recent years, and the use of FHB-resistant genotypes has been chosen as one of the prioritized trends in breeding practice [121,128,129]. Even now, however, the knowledge of complex mechanisms regulating resistance in cereal crops is still insufficient, and selecting resistant genotypes remains a difficult task for breeders. It has been established that, in addition to their fungicidal properties, a number of antioxidant secondary metabolites in cereals can regulate mycotoxin production by various pathogenic fungi [127].

The first weighty general argument in favor of phenolic compounds, carotenoids, and tocopherols is their ability to suppress reactive oxygen species (ROS), thus protecting biological cells. Besides, tocopherols and carotenoids can entrap free radicals of lipid peroxides and, therefore, arrest lipid peroxidation chain development [130]. Cinnamic acid derivatives, such as sinapic, caffeic, *p*-coumaric, chlorogenic, and ferulic acids, are effective inhibitors of *F. graminearum* and *F. culmorum* development, while benzoic acid derivatives, except syringic acid, produce an antiactivating effect [131,132]. There is an opinion that cereal crop metabolites with antioxidant activity suppress toxigenic action of a fungal infection. Numerous research works demonstrated the efficiency of phenolic compounds [133,134], carotenoids [135], tocopherols, and even benzoxazinoids [136] in restraining the growth and mycotoxin production of toxigenic *Fusarium* fungi. Finally, phenolic compounds partaking in plant structure enforcement are known to contribute to building a physical barrier against pathogenic infection. There is a positive interrelation between the content of phenolic acids, both free and bound to the cell wall, and FHB resistance in wheat [137]. A high level of FHB resistance in barley with the black-pigmented grain is supposedly associated with increased content of phenolic compounds [133].

High-protein oat forms were observed to be less affected by *Fusarium* head blight and accumulate fewer toxins; they are more adaptable to biotic stress. A relationship was identified between FHB resistance and accumulation of pipecolic acid, monoacylglycerols, tyrosine, galactinol, certain phytosterols, saccharides, and adenosine [121].

There were, however, many unproven assumptions on the participation of metabolites in the FHB resistance mechanism in cereals. Although the genetic architecture that supports secondary metabolite synthesis and regulation in cereal crops is exceptionally intricate, such proof may be retrieved in the process of comprehensive genetic and functional genomic studies [127].

Accumulation of avenanthramides in oats is also associated with the penetration of a fungal infection. Avenanthramides are mostly contained in oat grain, but under an attack by crown rust or leaf blotch, they start to synthesize in leaves as a means of protection against disease agents [110]. The fact that the amount of avenanthramides in grain significantly increases during imbibition [138], plant development [139], steeping [140], and storage [141] is also related to plant protection against potential susceptibility to pathogenic flora.

### 4.2. Abiotic Stress Resistance

Polyphenolic compounds in grain may protect seeds from unfavorable abiotic environmental conditions. Some of these compounds may act as sunscreens against potentially damaging UV-B radiation [142]. This may explain the presence of a purple grain color and other parts of the plant in tetraploid wheat *T. aethiopicum* Jakubz. [143] adapted to intensive solar UV-B radiation in highland areas in Ethiopia. Studies of near-isogenic wheat lines differing in the anthocyanin content in the pericarp and coleoptile under various stress conditions showed that both pericarp and coleoptile anthocyanins protected seedlings from osmotic stress [144], while protection of seedlings under a moderate irradiation dose (pretreatment of dry seeds with 50 Gy before sowing) or moderate Cd toxicity (25 µM CdCl_2_) was due to the coleoptile’s anthocyanins only [145,146]. Flavonoid substances can prevent negative effects of excessive moisture, such as pre-harvest seed sprouting by reducing the permeability of seed coat to water [147], inhibiting α-amylase (an enzyme whose activity is directly related to seed germination of grain) [148], or inactivating dehydrogenase required for the initial phase of respiration in ripening grain and young shoots [149].

Avenanthramide accumulation in oat grain is affected by weather and geographic conditions under which the studied material is cultivated [109,150,151,152,153]. Changes in the concentration of avenanthramides in response to salinity stress in CBF3 transgenic oat demonstrated that these compounds might have a potential role in enhancing abiotic stress tolerance in oats [154]. Havrlentova et al. [155] suggested that oats with higher β-D-glucan content may have thicker and, therefore, more insulating cell walls, better adapted to heat stress conditions. The same conclusion between higher content of β-D-glucan and greater cell wall thickness has been reported in barley [156]. Sterol might be important for cold acclimation of wheat [157,158] and oat [159]. Thus, breeding programs aimed at an increase in the content of health benefit components in cereal grain are at the same time eligible to solve the task of cultivar adaptability to unfavorable environmental conditions.

## 5. Conclusions

Each of the abovementioned natural components (dietary or curative) is promising for use as a food additive or an ingredient of pharmaceutical and cosmetic products. They are expected to play an ever-growing role in food industries, expanding the assortment of healthy food for the population. The demand for such products has already instigated plant breeders to launch new breeding programs aimed at the development of cereal crop cultivars with higher contents of bioactive components in grain. Such programs have often been based on molecular breeding techniques from the very beginning. Screening promising cultivars and hybrids for the content of antioxidants and other bioactive compounds in the grain is required to expand and promote this breeding trend. It also seems expedient to apply simple, undamaging and, as a rule, indirect techniques of plant genotype assessment for the levels of antioxidants in the grain to increase the performance and efficiency of such screening, employing the entire genetic diversity of cereal crops for identification of contrasting initial sources for breeding food and feed cultivars. The results obtained in the process of studying already existing cereal cultivars and the achievements of plant breeding in releasing new high-yielding and high-quality cultivars enable producers to use them in the development of a wide assortment of health-friendly dietary products contributing to the physical fitness of the human organism.

## Figures and Tables

**Table 1 plants-10-00086-t001:** Average values and ranges for the content of micronutrients (Fe, Mn, Zn) in caryopses of cereal crops [3].

Crops	Content, mg/kg		
	Fe	Mn	Zn
Winter soft wheat (*Triticum aestivum* L.)	**21.8** (19−4)	**4.3** (3.3−4.9)	**17.1** (13−21)
Spring soft wheat (*T. aestivum*)	**17.5** (15−22)	**3.3** (2.4−4.1)	**19.2** (14−22)
Soft wheat (mean)	**19.7** (15−24)	**3.8** (2.4−4.9)	**18.2** (13−22)
Winter and spring rye (*Secale cereale* L.)	**20.3** (14−30)	**4.2** (2.6−7.0)	**18.4** (15−24)
Spring barley (*Hordeum vulgare* L.)	**33.2** (24−79)	**10.1** (7−21)	**10.6** (6−33)
Oats (*Avena sativa* L.)	**26.7** (19−37)	**6.1** (3.5−9.9)	**26.3** (10−70)

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
