# Peer review of "Wheat, Barley, and Oat Breeding for Health Benefit Components in Grain"

_plants, 2021, doi:10.3390/plants10010086_

Round 1

Reviewer 1 Report

Abstract is too long. First sentence is not needed.

Line 17. Importance of starch in barley and oats? Should be explained in more details in the paper.

Line 63. Not all cereals are rich in proteins. Some are poor protein sources, in comparison to legumes.

Line 68. In many cases, high yielding ability is in negative correlation with content of nutritionally important substances.

Line 72. Some of genetic editing methods are not suitable for crops intended for ecological/environmental friendly growing, especially in the EU countries.

Lines 165-166. This is just a rough estimation, not enough supported by references in this paper.

Line 168. Other literature sources show essential difference in beta-glucan content from year to year. It is not enough just to make citation to only one literature source.

Line 176-180. More attention should be given to differences among grain tissues, and among milling fractions in this paper, not only in regard to beta-glucans, but as well about antioxidants, mineral elements, proteins etc.

Line 514. Breeding program should include the methods to alter the grain structures (for example multiple layer aleurones) as a way to improve the composition of cereals.

As well milling fractions with diverse content of beta-glucans in barley, oats etc should be taken into concern.

Author Response

To Reviewer 1

We very much appreciate the efforts of the reviewer to make our manuscript better. Our response to the reviewer’s comments can be found below.

When answering questions, adding references to literature, and moving parts of the text, the original line numbers have changed.

Abstract is too long. First sentence is not needed.

-        A: First sentence is deleted. Abstract is about 300 words, it is normal

Line 17. Importance of starch in barley and oats? Should be explained in more details in the paper.

  • - A: Information about the high starch content in barley and oats, its quality and use has been discussed many times in various publications. On the other hand, these crops are not starchy, like potatoes, and therefore selection for this indicator is not purposefully carried out, except for malting barley.

Line 63. Not all cereals are rich in proteins. Some are poor protein sources, in comparison to legumes.

-        A: Yes, you are absolutely right. It is changed. Wheat barley and oat are rich protein.

Line 68. In many cases, high yielding ability is in negative correlation with content of nutritionally important substances.

  • A: Yes, you are absolutely right. It is main purpose of breeding program.

Line 72. Some of genetic editing methods are not suitable for crops intended for ecological/environmental friendly growing, especially in the EU countries.

  • A: Genetic editing technologies are new innovative directions in plant breeding. There are many questions about these technologies, but they demonstrate practical results that we have never obtained using conventional breeding methods. We do not consider legislative issues in this article.

Lines 165-166. This is just a rough estimation, not enough supported by references in this paper.

  • A: Up to 166 line in this section concerning β-glucans, more than 24 publications are cited.

Line 168. Other literature sources show essential difference in beta-glucan content from year to year. It is not enough just to make citation to only one literature source.

  • A: About 40 publications are cited in this section on β-glucans.

Line 176-180. More attention should be given to differences among grain tissues, and among milling fractions in this paper, not only in regard to beta-glucans, but as well about antioxidants, mineral elements, proteins etc.

  • A: Unfortunately, it focused here primarily on genetic and breeding issues, of which there are a lot, and we are trying to identify the sources of all these Health Benefit Components, with which can conduct in-depth study of technical issues of processing in further.

Line 514. Breeding program should include the methods to alter the grain structures (for example multiple layer aleurones) as a way to improve the composition of cereals.

  • A: Unfortunately, we will never be able to get such a result by conventional selection methods. For structural changes in caryopsis, it is necessary to know the quantity, quality and heritability of these complex Health Benefit Components, which is what this publication is about. And only then, on the basis of molecular genetic studies using genetic editing techniques or similar technologies, can one try to obtain such a result.
  •  

As well milling fractions with diverse content of beta-glucans in barley, oats etc should be taken into concern.

  • A: Yes, you are absolutely right. Unfortunately, the article cannot be endless.

Reviewer 2 Report

Thank you for the very nice paper, and I enjoyed and learnt a lot whilst reading it. Some comments:

Lines 78-124: No discussion about wheat ?

Line 93: Add other references that combine both breeding and agrochemical techniques to increase nutrient content.

Line 96: Add reference

Line 101: Which countries do these cultivars come from ?

Line 109-110: Where do these oat varieties come from ?

Lines 128-129: I would not say "typical of all Poaceae Family" because rice and millet, for example, are members of the Poaceae but they have non-detectable beta-glucan content.

Line 132: "As opined by many scientists" - where are the references ?

Line 176: Which polysaccharide ?

Lines 136-175 (and also check the whole document): Needs rewriting/restructuring because the paragraphs tend to jump between oat and barley. Write about oat first (or barley, if you prefer), then move to the next crop in the next paragraph. Do not mix both in the same paragraph since it is confusing to follow. The same comment for the other sections.

Line 181: How about starch ? Starch can also affect viscosity.

Line 189: What about insoluble dietary fibre ? What is its importance in relation to soluble dietary fibre and health ?

Line 197: What type of mutagenesis ?

Line 235: What does it mean by "provocative conditions" ?

Lines 249-251: Folic acid is not a phenolic acid but it is included under the heading "Phenolic acids".

Line 279: What is the Nud gene and what is its function ?

Line 316: Antiprolife ? I have not seen this word before in English...

Lines 337,338: Spell out "4".

Lines 341-343: I would put it after "α, β, γ and δ [131]." in line 331.

Line 347: Under "Sterol" section - what about discussion on wheat and barley ?

Line 358: How much carotenoids in wheat and barley ? Maybe add a discussion on carotenoids in wheat and barley.

Lines 448,454: FHB was defined twice.

Line 493: Why would the phenolics accummulate only in the grain, and not in the whole plant, which is also exposed to UV ?

Line 551: By what mechanism does the β-D-glucan content increase tolerance to heat stress ?

Under Section: "The Effect of Dietary Components in Grain on Life Functions of Plants Themselves", does the plant develop these bioactives after pathogenic attack, or is it innate in the plant (meaning, does the plant already have high levels of bioactives prior to pathogen attack ?)

Author Response

To Reviewer 2

We very much appreciate the efforts of the reviewer to make our manuscript better. Our response to the reviewer’s comments can be found below.

When answering questions, adding references to literature, and moving parts of the text, the original line numbers have changed.

Lines 78-124: No discussion about wheat?

  • A: Wheat is mentioned on 88, 99, 102 lines

Line 93: Add other references that combine both breeding and agrochemical techniques to increase nutrient content.

  • A: It is added

Line 96: Add reference

  • A: It is added

Line 101: Which countries do these cultivars come from?

  • A: Russia

Line 109-110: Where do these oat varieties come from?

  • A: Different geographical origin

Lines 128-129: I would not say "typical of all Poaceae Family" because rice and millet, for example, are members of the Poaceae but they have non-detectable beta-glucan content.

  • A: Typical of some species Poaceae Family

Line 132: "As opined by many scientists" - where are the references?

  • A: It is added

Line 176: Which polysaccharide?

  • A: β-glucans

Lines 136-175 (and also check the whole document): Needs rewriting/restructuring because the paragraphs tend to jump between oat and barley. Write about oat first (or barley, if you prefer), then move to the next crop in the next paragraph. Do not mix both in the same paragraph since it is confusing to follow. The same comment for the other sections.

  • A: In most cases, β-glucans were studied in barley and oats together or in similar groups of varieties (naked and hulled) and these parameters behaved in a similar way, therefore we consider them in parallel.

Line 181: How about starch? Starch can also affect viscosity.

  • A: Yes, β-glucan, being non-starch polysaccharide, has a viscosity effect.

Line 189: What about insoluble dietary fibre? What is its importance in relation to soluble dietary fibre and health?

  • A: Insoluble fractions of dietary fibre are glucose (cellulose), xylose and arabinose. β-glucans is the main component of the soluble fibre. Insoluble dietary fibre has general gastrointestinal effects and is used in most cases for lose weight [51, 102].

Line 197: What type of mutagenesis?

  • A: Mutations induced by TILLING of high frequency mutagenesis with molecular based

Line 235: What does it mean by "provocative conditions"?

  • A: mold-friendly conditions (increased moisture and temperature 23-25C)

Lines 249-251: Folic acid is not a phenolic acid but it is included under the heading "Phenolic acids".

  • A: You are absolutely right. Folic acid has general antioxidant properties

Line 279: What is the Nud gene and what is its function?

  • A: It is added. It is gene of nakedness

Line 316: Antiprolife? I have not seen this word before in English...

  • A: Sorry, it is mistake only, of course, it is antiproliferative

Lines 337,338: Spell out "4".

  • A: It is four

Lines 341-343: I would put it after "α, β, γ and δ [131]." in line 331.

  • A: Sorry, not clear what needs to be done, but it is changed.

Line 347: Under "Sterol" section - what about discussion on wheat and barley?

  • A: It is added

Line 358: How much carotenoids in wheat and barley? Maybe add a discussion on carotenoids in wheat and barley.

  • A: Information about carotenoids in wheat and barley was in the text

Lines 448,454: FHB was defined twice.

  • A: Fusarium Head Blight

Line 493: Why would the phenolics accummulate only in the grain, and not in the whole plant, which is also exposed to UV?

  • A: Presence of purple grain color and other parts of the plant in tetraploid wheat adapted to intensive solar UV-B radiation in highland areas in Ethiopia

Line 551: By what mechanism does the β-D-glucan content increase tolerance to heat stress?

  • A: It was suggested that oats with higher β-D-glucan content may have thicker and therefore more insulating cell walls, better adapted to heat stress conditions

Under Section: "The Effect of Dietary Components in Grain on Life Functions of Plants Themselves", does the plant develop these bioactives after pathogenic attack, or is it innate in the plant (meaning, does the plant already have high levels of bioactives prior to pathogen attack?)

  • A: Content of biochemical components in cereals crops there are variations in the composition of these products. These variations arise from differences between environments, variation in the genotype of the crop and from interactions biotic and abiotic factors and genotype.

Reviewer 3 Report

General comments:

The main objective of this review paper was to discuss new breeding programs to developed new cereal grain richer in bioactive compounds, vitamins, dietary fibres and oils, etc., - while growing plants -, and based on plant genetic resources deposited in gene banks such as landraces, rare crop species or even wild relatives of cultivated plants. As regards with this new genotypes, the resistance to diseases or tolerance to certain abiotic stressors is also discussed because these cereal grains might at the same time match the task of developing cultivars adapted to unfavourable environmental conditions.

The topic is of interest because for a long time cereal grains have been overall selected more for their high bread making quality and protein content, more based on economic objective than health issues. The new tendency to shift selection towards cereal grains for higher density in health protective bioactive compounds and dietary values deserves to be addressed and underlined.

The review is correctly written. However, the structuration of the review somewhat appears confused and would deserve a clearer organization in the arguments, e.g.:

  • From line 126 about “beta-glucans” and “antioxidants”: there is a lack of hierarchization of the ideas which general considerations, health issues, and genetic/breeding issues mixed together without a clear organization: these sections have to be restructured. For example, I will propose:
    • Description of the health properties of the compounds to justify to increase its content
    • What’s about its presence in commercial and other varieties (local, wild…)
  • The other two chapters should be re-organized more clearly, i.e.:
    • “Assessment of Cereal Crop Genetic Resources According to the Diversity and Concentration of Health-Friendly Dietary Grain Components”
    • And “The Effect of Dietary Components in Grain on Life Functions of Plants Themselves”
  • Indeed, in these sections, you only partially addressed the main objective of you review, i.e., comparisons between commercial varieties and others… The distinctions were not clear.

Other generic remarks:

  • At several places, it lacks references to address your discussion and assertions;
  • It lacks illustrations, either at least one Table or one Figure;
  • Nothing about organic cereal grains as a way to increase nutrient density?
  • An important issue not discussed here: cereal processing. Indeed, you can have micronutrient-dense varieties, but if processing is too drastic, this will useless because refining remove more than 80% of micronutrients and fibre, and extrusion-cooking may lead to very high glycaemic indices. Therefore, the issue of nutrient-dense varieties must be coupled with minimal processing to be relevant on long term. This point should be addressed, at least in conclusion.

Overall, the advantages of new varieties, i.e., wild, ancient, landraces, and/or rare crop/varieties, does not appear very clearly in your review. The structure of the review should be more built on recent versus ancient varieties. For example, lines 514-516, you only concluded: “Thus, breeding programs aimed at an increase in the content of health benefit components in cereal grain are at the same time eligible to solve the task of cultivar adaptability to unfavorable environmental conditions.” Besides, in the “Biotic Stress Resistance” section, there is no clear mention of the advantages of ancient or wild varieties.

Specific comments:

Abstract: well written and clear

Line 40: “consumed by humans”, add a reference please.

Line 54: you should cite them, e.g., antioxidants, lipotropic compounds…

Line 63: please add references

Lines 74-75: why this focus on wheat, barley and oat breeding? Please justify. Why not addressing maize and/or rice?

Line 81: “poor in both the amount of micronutrients”: due to breeding or refining? Please clarify, this is an important point.

Lines 81-88: Yes, but this may be due also to their monotonous diets, not only the nutrient density of cereals: this should be mentioned.

Lines 94-99: A table with comparisons in micronutrient contents of commercial cultivars, local varieties, etc., will be relevant here to give an idea of the difference in ranges. Are standard deviations larger for ancient/local varieties than with commercial ones?

Line 115: please add a reference

Line 118: please add a reference

Lines 118-124: What did you mean by “in this work” and “in the course of the present study”? This is unclear.

Lines 131-134: please add references

Lines 176-195: you should move upward this paragraph, at the beginning of the section to give an argument (or the basis) for increasing its content in cereal grains. Indeed, just after this section about health potential you move towards genetic considerations.

Lines 223-231: Yes, but there is the issue of antioxidant bioavailability in humans, which can be low. For example, around 95% of phenolic antioxidants are linked to fibre.

Lines 236-237: also some trace elements and minerals that may act as antioxidant enzyme co-factors in humans.

Lines 248-251: this section is about “Phenolic Compounds and Avenanthramides”: why mentioning folic acid?

Line 255: please add a reference

Lines 252-259: this section should be at the beginning of the section

Lines 260-266: why addressing “melanin”? I don’t understand.

Line 328: “phenolics and sterines”: they are not tocols?

Line 392: “whose content reduced”: it lacks “is”?

Lines 423-427: the sentence is rather confused: please, re-write

Lines 428-429: I don’t understand why you are mentioning “plant material modifiers, chemical additives or filling agents are often used for desired end-product quality improvement”? What is the link with what follows below?

Lines 514-516: this section should be in the Conclusion section.

Lines 517-519: what is the relation between this paragraph and the main topic of the section? I don’t understand.

Lines 522-523: What is the link with the main topic of the review? The objective is rather to limit the use of such additives via developing nutrient-dense new cereal grain varieties?

Line 525: “Product” not “produce”?

Lines 534-537: provided food processing does not level the nutrient density? An issue to address.

Author Response

To Reviewer 3

We very much appreciate the efforts of the reviewer to make our manuscript better. Our response to the reviewer’s comments can be found below.

The review is correctly written. However, the structuration of the review somewhat appears confused and would deserve a clearer organization in the arguments, e.g.:

From line 126 about “beta-glucans” and “antioxidants”: there is a lack of hierarchization of the ideas which general considerations, health issues, and genetic/breeding issues mixed together without a clear organization: these sections have to be restructured. For example, I will propose: Description of the health properties of the compounds to justify to increase its content

What’s about its presence in commercial and other varieties (local, wild…)

  • Some paragraphs were moved to the beginning of sections for clearer organization in the arguments

The other two chapters should be re-organized more clearly, i.e.: “Assessment of Cereal Crop Genetic Resources According to the Diversity and Concentration of Health-Friendly Dietary Grain Components”

And “The Effect of Dietary Components in Grain on Life Functions of Plants Themselves”

Indeed, in these sections, you only partially addressed the main objective of you review, i.e., comparisons between commercial varieties and others… The distinctions were not clear.

Other generic remarks:

At several places, it lacks references to address your discussion and assertions;

It lacks illustrations, either at least one Table or one Figure;

-        A: Table is added

Nothing about organic cereal grains as a way to increase nutrient density?

An important issue not discussed here: cereal processing. Indeed, you can have micronutrient-dense varieties, but if processing is too drastic, this will useless because refining remove more than 80% of micronutrients and fibre, and extrusion-cooking may lead to very high glycaemic indices. Therefore, the issue of nutrient-dense varieties must be coupled with minimal processing to be relevant on long term. This point should be addressed, at least in conclusion.

  • A: Unfortunately, the aim of the current review is to consider current trends and achievements in wheat, barley and oat breeding for health-benefiting components. It is not discussed grain processing issues here. This topic is quite large and complex and is not part of the scope of this review.

Overall, the advantages of new varieties, i.e., wild, ancient, landraces, and/or rare crop/varieties, does not appear very clearly in your review. The structure of the review should be more built on recent versus ancient varieties. For example, lines 514-516, you only concluded: “Thus, breeding programs aimed at an increase in the content of health benefit components in cereal grain are at the same time eligible to solve the task of cultivar adaptability to unfavorable environmental conditions.” Besides, in the “Biotic Stress Resistance” section, there is no clear mention of the advantages of ancient or wild varieties.

  • A: Unfortunately, the systematic study of a wide range of qualitative biochemical characteristics of landraces, old varieties or wild relatives of cultivated plants is not very numerous for grain crops.

Specific comments:

When answering questions, adding references to literature, and moving parts of the text, the original line numbers have changed.

Line 40: “consumed by humans”, add a reference please.

  • A: It is added

Line 54: you should cite them, e.g., antioxidants, lipotropic compounds…

  • A: It is added

Line 63: please add references

  • A: It is added

Lines 74-75: why this focus on wheat, barley and oat breeding? Please justify. Why not addressing maize and/or rice?

  • A: The article cannot be endless and cover all crops at once, restrictions are indicated in the title of the article.

Line 81: “poor in both the amount of micronutrients”: due to breeding or refining? Please clarify, this is an important point.

  • A: It is added. Until recently the breeding of these crops was aimed at obtaining the highest productivity, often at the expense of the quality of the products obtained.

Lines 81-88: Yes, but this may be due also to their monotonous diets, not only the nutrient density of cereals: this should be mentioned.

  • A: It is added. Yes, you are absolutely right, serious global problem invoked by the uniformity different diets and may lead to significant health deteriorations

Lines 94-99: A table with comparisons in micronutrient contents of commercial cultivars, local varieties, etc., will be relevant here to give an idea of the difference in ranges. Are standard deviations larger for ancient/local varieties than with commercial ones?

  • A: It is added.

Line 115: please add a reference

  • A: It is added.

Line 118: please add a reference

  • A: It is added.

Lines 118-124: What did you mean by “in this work” and “in the course of the present study”? This is unclear.

  • A: It was a mistake

Lines 131-134: please add references

  • A: It is added.

Lines 176-195: you should move upward this paragraph, at the beginning of the section to give an argument (or the basis) for increasing its content in cereal grains. Indeed, just after this section about health potential you move towards genetic considerations.

  • A: The paragraph was moved to the beginning of the section.

Lines 223-231: Yes, but there is the issue of antioxidant bioavailability in humans, which can be low. For example, around 95% of phenolic antioxidants are linked to fibre.

  • A: Unfortunately, medical concerns are not addressed here antioxidant bioavailability in humans

Lines 236-237: also some trace elements and minerals that may act as antioxidant enzyme co-factors in humans.

  • A: You are absolutely right, but it focused on the problems of the availability of dietary components, their genetics and breeding use.

Lines 248-251: this section is about “Phenolic Compounds and Avenanthramides”: why mentioning folic acid?

  • A: It was a mistake, it was moved

Line 255: please add a reference

  • A: It is added.

Lines 252-259: this section should be at the beginning of the section

  • A: The paragraph was moved to the beginning of the section.

Lines 260-266: why addressing “melanin”? I don’t understand.

  • A: The chapter describes phenolic compounds and their antioxidant properties, therefore it seemed appropriated for us to mention seed melanins known to be strong antioxidants among other phenolic antioxidants. To clarify this we improved the sentence: «In the study of molecular mechanisms of ‘melanin-like’ black seed pigments known to be strong antioxidants….”

Line 328: “phenolics and sterines”: they are not tocols?

  • A: You are absolutely right. It was a mistake
  •  

Line 392: “whose content reduced”: it lacks “is”?

  • A: Metabolites, which content reduced in the process of domestication or in which wild oat species differed from oat cultivars, is identified

Lines 423-427: the sentence is rather confused: please, re-write

  • A: It is divided into two sentences

Lines 428-429: I don’t understand why you are mentioning “plant material modifiers, chemical additives or filling agents are often used for desired end-product quality improvement”? What is the link with what follows below?

  • A: It was changed sentence

Lines 514-516: this section should be in the Conclusion section.

-        A: The paragraph was moved in the Conclusion

Lines 517-519: what is the relation between this paragraph and the main topic of the section? I don’t understand.

  • A: The paragraph was moved to another section

Lines 522-523: What is the link with the main topic of the review? The objective is rather to limit the use of such additives via developing nutrient-dense new cereal grain varieties?

  • A: It said about abovementioned natural components – fibers, phenolic acids, tocols, avenanthramide etc. in new developed cultivars

Line 525: “Product” not “produce”?

  • A: You are absolutely right. It was a mistake.

Lines 534-537: provided food processing does not level the nutrient density? An issue to address.

  • A: You are absolutely right. It focused here primarily on genetic and breeding issues, of which there are a lot, and we are trying to identify the sources of all these Health Benefit Components, with which can conduct in-depth study of technical issues of processing in further.

Round 2

Reviewer 2 Report

The quality is greatly improved. Well done. I like your paper.

Just a minor comment and it will be ready for publication:

You only have single sentences in some "paragraphs" (see below). Paragraphs cannot contain just one sentence. You need at least two sentences to form a paragraph. You could potentially combine these sentences with the preceding paragraph or the one succeeding it, without ruining the flow of the ideas or thoughts.

Lines 207-210

Lines 211-214

Lines 227-229

Lines 240-243; 244-245

Lines 252-253

Lines 275-277

Lines 323-324

Lines 342-344

Lines 352-354

Lines 534-536

Author Response

Dear Reviewer,

We very much appreciate the efforts of the reviewer to make our manuscript better ones more. We are combined all sentences in your mentioned lines. Thank you so much.

Reviewer 3 Report

All concerns have been adequately adressed with substantial changes in the structuration of the manuscript. References and a Table have been added. However, for a next paper, think to indicate changes made in red, and give the new pages and lines for the changes.

  What do you want to do ? New mailCop

Author Response

Dear Reviewer,

We very much appreciate the efforts of the reviewer to make our manuscript better ones more. We made some changes and used all your recommendations and will take them into account when writing and editing the next article. Thank you very much.